# Hyperhomocysteinemia and Cardiovascular Disease: Is the Adenosinergic System the Missing Link?

**DOI:** 10.3390/ijms22041690

**Published:** 2021-02-08

**Authors:** Franck Paganelli, Giovanna Mottola, Julien Fromonot, Marion Marlinge, Pierre Deharo, Régis Guieu, Jean Ruf

**Affiliations:** 1C2VN, INSERM, INRAE, Aix-Marseille University, F-13005 Marseille, France; franck.paganelli@ap-hm.fr (F.P.); giovanna.mottola@univ-amu.fr (G.M.); julien.fromonot@univ-amu.fr (J.F.); marion.marlinge@ap-hm.fr (M.M.); pierre.deharo@ap-hm.fr (P.D.); guieu.regis@orange.fr (R.G.); 2Department of Cardiology, North Hospital, F-13015 Marseille, France; 3Laboratory of Biochemistry, Timone Hospital, F-13005 Marseille, France; 4Department of Cardiology, Timone Hospital, F-13005 Marseille, France

**Keywords:** adenosine, adenosine A_2A_ receptors, cardiovascular disease, hydrogen sulfide, hyperhomocysteinemia

## Abstract

The influence of hyperhomocysteinemia (HHCy) on cardiovascular disease (CVD) remains unclear. HHCy is associated with inflammation and atherosclerosis, and it is an independent risk factor for CVD, stroke and myocardial infarction. However, homocysteine (HCy)-lowering therapy does not affect the inflammatory state of CVD patients, and it has little influence on cardiovascular risk. The HCy degradation product hydrogen sulfide (H_2_S) is a cardioprotector. Previous research proposed a positive role of H_2_S in the cardiovascular system, and we discuss some recent data suggesting that HHCy worsens CVD by increasing the production of H_2_S, which decreases the expression of adenosine A_2A_ receptors on the surface of immune and cardiovascular cells to cause inflammation and ischemia, respectively.

## 1. Introduction

Cardiovascular disease (CVD) includes conditions that affect the heart or blood vessels. The heart is a pump, and blood vessels are conduits for blood and cells that supply oxygen and nutrients to maintain the molecular mechanisms necessary for vascular development and the functioning of different tissues. Each organ has its own capillary network to fulfill its specific functions, and endothelial cells provide the microvasculature of the different organs. These endothelial cells form a vascular wall that controls organ development, homeostasis and tissue regeneration. Pathological processes, such as arteriosclerosis, compromise the integrity and structure of this vascular wall, and arteriosclerosis most often leads to CVD. The initiating events of atherogenesis involve the retention of lipoproteins in the subendothelial space of the arteries and the activation of endothelial cells. Circulating monocytes adhere to activate endothelial cells, enter the vascular wall, and differentiate into tissue macrophages. These macrophages ingest lipoproteins and turn into foam cells. In addition, synthetic vascular smooth muscle cells accumulate in atheromas and secrete extracellular matrix proteins, and smooth muscle cells and collagen are important components of the fibrous cap that covers the atherosclerotic plaque. It is believed that plaques with a reduced ratio of smooth muscle cells to foam cells are vulnerable to rupture, which is the event inducing thrombosis and, therefore, myocardial infarction [1].

CVD is a leading cause of death, but how the multifactorial pathology develops is not clear. The incidence of cardiovascular morbi-mortality from CVD varies according to conventional risk factors [2]. Factors that affect the risk of developing CVD include a genetic history (gender, family, or ethnicity) [3] or a poor lifestyle (smoking, alcohol use, lack of activity, or unhealthy diet) [4]. Hypertension is the most common modifiable risk factor in CVD [5]. High blood pressure is often associated with metabolic deregulation, which leads to high blood cholesterol levels that, such as glucose in type 2 diabetes, damage blood vessels and lead to atherosclerosis. The mechanisms that link the regulation of blood pressure and hypercholesterolemia, the mutual interaction between hypertension and hypercholesterolemia and their influence on the development of atherosclerosis are mainly the renin-angiotensin-aldosterone system, oxidative stress, endothelial dysfunction and increased production of endothelin-1 [6]. Hypertension is also associated with metabolic deregulation of the methionine cycle, which leads to hyperhomocysteinemia (HHCy) [7]. Of all the established risk factors associated with the development of hypertension and its complications such as accelerated cardiac atherosclerosis and premature death, HHCy is probably the most elusive. The aim of this review is to propose a mechanism by which HHCy causes an increase in H_2_S levels, which affects the adenosinergic system, ultimately promoting CVD.

## 2. HHCy as a Risk Factor in CVD

Homocysteine (HCy) is a thiol group-containing amino acid metabolite that is produced in all cells via the methionine cycle. HCy synthesis occurs via the transmethylation of methionine by S-adenosylmethionine synthetase (SAMS) to form SAM from methionine and ATP. Methyltransferase (MT) converts SAM to S-adenosylhomocysteine (SAH), and SAH hydrolase metabolizes SAH into adenosine and HCy [8,9,10] (Figure 1). HCy is remethylated to methionine and transsulfurated to cysteine. Remethylation of HCy involves folate/vitamin B_12_-dependent and vitamin B_12_-independent mechanisms. The former step is catalyzed by the vitamin B_12_-dependent enzyme methionine synthase (MS) and uses *N*-5-methyl tetrahydrofolate (THF) as the methyl group donor, and the latter step is catalyzed by betaine-homocysteine S-methyl transferase (BHMT) and uses the methyl group from betaine [11] (Figure 1).

Approximately 5~10% of the total daily cellular production of HCy that is not metabolized within the cell is exported to the plasma compartment, where normal HCy levels range from 5 to 15 μmol/L, and this baseline value is maintained in healthy human subjects via constant clearance by the kidney [12,13,14]. Vitamin B_12_ and folic acid deficiencies may lead to HHCy, which is linked to the development of CVD [15,16]. HHCy is a condition in which the plasma concentration of HCy is elevated, which occurs as a result of an imbalance between its biosynthesis and catabolism [17]. The definition of HHCy is controversial, but it is generally defined as plasma HCy ≥ 10 μmol/L [18,19,20]. However, a slight increase (10–15 μmol/L) in plasma HCy level is associated with morbi-mortality [21], and a higher cut-off (≥15 μmol/L) was also considered to designate HHCy [22,23]. In conclusion, HHCy is categorized into three classes as mild, moderate and severe HHCy with plasma HCy levels ranging from 15 to 30 μmol/L, 31 to 100 μmol/L and > 100 μmol/L, respectively [24].

HCy contributes to the development of CVD via several mechanisms, such as its adverse effects on the vascular endothelium and smooth muscle cells, which lead to alterations in subclinical arterial structure and function. Therefore, HHCy is an independent risk factor for atherosclerosis leading to CVD [16,25,26,27]. Several studies showed a clear correlation between HCy plasma levels and the severity of atherosclerosis [28] and support an association between elevated HCy levels and increased cardiovascular mortality [29]. HHCy is associated with the etiology of myocardial infarction and stroke, but the mechanisms of HCy promotion of CVD are not clear [30,31]. HCy may promote CVD via mechanisms involving vascular muscle cell proliferation, a decrease in circulating HDL, conversion to HCy-thiolactone and induction of an autoimmune response and thrombogenesis [32,33,34,35]. HHCy activates Nuclear Factor-kappa B (NF-κB), which regulates the transcription of various genes involved in inflammatory and immune responses to increase pro-inflammatory cytokines and downregulate anti-inflammatory cytokines [36]. HHCy also induces endothelial cell dysfunction by decreasing endothelial antioxidant defense to cause oxidative stress and an increase in the intracellular concentration of reactive oxygen species (ROS) [37]. ROS disturb lipoprotein metabolism, which contributes to the growth of atherosclerotic vascular lesions [38]. HCy acts on vessels by controlling the contractility of vascular smooth muscle cells and the permeability of endothelial cells via the inhibition of endothelial nitric oxide synthase, which produces nitric oxide (NO) [39,40]. Increased HCy is also associated with DNA hypomethylation in vascular disease [41] but this complex regulatory mechanism is tissue-specific [42]. To alleviate the intracellular accumulation of HCy when the remethylation pathway is impaired, endothelial cells export HCy to the circulation [43]. The mechanism of HCy transport in the vascular endothelium is not well defined, but human aortic endothelial cells bind and import l-HCy via at least four of the known cysteine sodium-dependent transport systems, namely, X_AG_, L, ASC and A, and l-homocysteine is imported via the X_AG_, L, ASC and xc systems [44]. The effects of HHCy on CVD may also be due to the increased production of hydrogen sulfide (H_2_S).

## 3. H_2_S as a Gasotransmitter

H_2_S is the final product of HCy metabolism. Transsulfuration of HCy is catalyzed via the vitamin B_6_-dependent enzymes cystathionine β-synthase (CBS) and cystathionine γ-lyase (CSE). CBS converts HCy and serine into cystathionine, which is used by CSE to generate cysteine [8,9] (Figure 2). CBS and CSE are the major enzymes responsible for the biogenesis of hydrogen sulfide, which is endogenously generated in mammalian tissues [45,46]. CBS and CSE catalyze various biochemical mechanisms. CBS produces H_2_S from cysteine via a β-elimination reaction, and CSE generates H_2_S via the α,β-elimination of cysteine. CBS and CSE perform a β-replacement reaction, which condenses two cysteine molecules or catalyzes the condensation reaction of HCy with cysteine via β- or γ-replacement to produce H_2_S. CBS and CSE also affect cysteine α,β-elimination production of cysteine persulfide to ultimately generate H_2_S [47] (Figure 2). CBS also produces H_2_S via β-replacement in which cysteine is hydrolyzed and condensed with HCy, which provides a biochemical explanation for the HCy-lowering effects of N-acetylcysteine treatments in humans [48]. Under conditions of high HCy levels, the α,γ-elimination and γ-replacement reactions likely account for most H_2_S production by CSE [49] (Figure 2). 3-Mercaptopyruvate (3-MP) sulfurtransferase (MST) in combination with cysteine aminotransferase (CAT) also produces H_2_S from cysteine [50,51] (Figure 2). l- and d-cysteine may be involved in the biosynthetic pathway for the production of H_2_S via MST and d-cysteine oxidase. The d-cysteine-dependent pathway acts primarily in the cerebellum and kidney [52]. Cardiovascular cells and tissues are limited in their capacity to metabolize HCy because these cells do not express cystathionine β-synthase, which is the first enzyme in the transsulfuration pathway [53,54].

Genetic disorders in the enzymes responsible for HCy metabolism, such as mutations in N-5,10-methylenetetrahydrofolate reductase (MTHFR) and CBS, may result in moderate or severe HHCy [55]. CBS, CSE and MST are differentially expressed in various systems and affect the functions of these systems via the production of H_2_S. The physiological functions of H_2_S are mediated via different molecular targets, such as different ion channels and signaling proteins [56]. Immunohistochemistry localized CBS in the endothelium of small pial arteries and intracerebral arterioles, capillary walls, neurons and vascular nerves. CSE is localized in smooth muscles and the thoracic aorta, where it is the main enzyme producing H_2_S. MST and CAT are localized in the vascular endothelium of the thoracic aorta [57,58].

H_2_S is a gasotransmitter known for its regulatory role in many physiological processes [59]. H_2_S works with NO and carbon monoxide (CO) as an important endogenous signaling molecule in mammalian cells and tissues [60,61,62,63,64], specifically in the cardiovascular and nervous systems [65,66]. One important property of H_2_S is its biphasic pharmacological mode of action. At low concentrations, H_2_S exerts modulatory effects and acts as a cytoprotective, antioxidant and anti-inflammatory agent [63,67]. In contrast, higher concentrations of H_2_S induce deleterious actions, including pro-oxidant effects and cytostatic and cytotoxic responses. These responses involve numerous signal transduction pathways and molecular targets, including K_ATP_ channels, Akt, AMP kinase, PTEN, NF-κB, Nrf2, proline-rich kinase 2, the adenylate cyclase and guanylate cyclase systems and inhibition of cytochrome C oxidase [63,67,68,69].

### 3.1. H_2_S in Pathophysiological Conditions

The role of H_2_S in vascular diseases, inflammation, critical illness, reperfusion injury, various nervous system diseases, metabolic diseases and cancer was extensively reviewed [70,71,72,73,74,75,76,77,78]. Evidence in support of a role of H_2_S deficiency in vascular disorders, such as hypertension and atherosclerosis, is accumulating [79,80]. Other review articles addressed the vascular biology of H_2_S and the mechanisms of HHCy-induced vascular injury [81,82]. CSE may be primarily responsible for changes in H_2_S production in HHCy [49]. Accumulating evidence supports the inhibitory effect of HCy on H_2_S generation [83,84,85], but there were also reports of elevated H_2_S levels in HHCy [86,87]. The evidence of defective and enhanced H_2_S production under HHCy conditions and the metabolic imbalance of HCy and H_2_S in cardiovascular pathologies suggests that changes in the H_2_S/HCy ratio may be more valuable than changes in the absolute concentrations of H_2_S and HCy in depicting the role of these metabolites in disease pathogenesis [88,89,90]. There are growing controversies on the physiologically significant concentrations of H_2_S and its biological effects [91,92]. H_2_S stimulates or inhibits intracellular transduction pathways, cell proliferation, apoptosis and hemostasis [59,93,94,95,96,97,98]. H_2_S also exerts pro- and anti-inflammatory effects [65,99,100]. Only a few studies showed changes in the levels of H_2_S in human diseases, and most of these measurements were indirect and measured compounds linked to H_2_S, such as thiosulfate or sulfhemoglobin, rather than H_2_S itself [101]. Due to its vasorelaxative and vasoprotective properties, H_2_S may be useful in the treatment of arterial hypertension by decreasing peripheral resistance [92]. Research on the clinical and fundamental aspects of H_2_S is in full development, particularly the relationship between the production of H_2_S and epigenetics such as DNA methylation and DNA damage repair [102].

### 3.2. H_2_S in Immune Cells

The immune cells are in permanent contact with H_2_S because of the endogenous and exogenous production from the surrounding parenchymal cells, which regulates their viability and function. The downregulation or genetic defect in endogenous H_2_S-producing enzymes leads to the onset or development of autoimmune diseases [103]. Monocytes/macrophages express CBS, CSE and MST, and CSE and MST are likely the primary enzymes. Pro-inflammatory agents, such as liposaccharide (LPS), tend to upregulate CSE, and anti-inflammatory and cytoprotective agents, such as steroids and statins, inhibit its expression [104,105,106,107]. Bacterial LPS increases CSE expression and concomitant H_2_S production in macrophages via activation of the p38 MAP kinase pathway, and glucocorticoids prevent this upregulation [104,105,108]. CSE upregulation in macrophages is also dependent on activation of the NF-κB and ERK pathways [109]. H_2_S-induced signaling appears to play an important role in T cell activation. Whether H_2_S is produced by activated T cells or administered exogenously, it acts as an autocrine or paracrine enhancer of T cell activation. Administration of H_2_S at nanomolar concentrations boosts T cells and upregulates the expression of activation markers, such as CD69, IL-2 and CD25 [110]. The action of H_2_S protects immune cells from various deleterious effects, such as oxidative stress (ROS production) or inflammatory runaway. The exogenous administration of H_2_S donors exerted anti-inflammatory effects in various local and systemic inflammatory diseases, such as brain disease, neoplastic disease of the colon, inflammatory disease joints, kidneys, cardiovascular, ophthalmic and dermatological diseases [67,111,112,113,114,115,116,117,118,119,120,121,122,123]. Paradoxically, a given H_2_S donor may have beneficial effects on the immune system but undesirable effects on the vascular system and vice versa. Although H_2_S acts on various immune cells (macrophages, T lymphocytes, etc.), it exerts little-known complex effects on the interactions between immune and nonimmune processes. For example, H_2_S reversed adenosinergic impairment of T cell viability via suppression of NF-κB, which downregulated A_2A_R expression and may also affect the cardiovascular system [124].

## 4. Adenosine as a Purinergic Modulator of Cardiovascular and Immune Systems

Adenosine is a ubiquitous autacoid that is derived from the dephosphorylation of ATP intracellularly or via the extracellular ectoenzymes CD39 and CD73. Most types of cells release ATP. At the intracellular level, some adenosine also originates from the methionine cycle via the hydrolysis of SAH, which leads to the formation of HCy and adenosine in a stoichiometric ratio. Intra- and extracellular adenosine is deaminated to inosine by adenosine deaminase and joins the end product of the catabolism of purines, uric acid via nucleosidase and xanthine oxidase to yield hypoxanthine and xanthine, respectively [125,126] (Figure 3). Extracellularly, adenosine acts as a signaling molecule by interacting with the integral membrane proteins adenosine receptors or P1 purinergic receptors. Four subtypes were cloned and named: A_1_, A_2A_, A_2B_ and A_3_ receptors. The intracellular segment of each adenosine receptor subtype interacts with the appropriate heterotrimeric guanine nucleotide-binding protein (G-protein) with subsequent activation of an intracellular signal transduction mechanism. Adenosine receptor subtypes are divided into two main categories: (i) receptors that are coupled to inhibitory G-proteins (Gi), such as adenosine A_1_R and A_3_R; and (ii) receptors that are coupled to stimulating G-proteins (Gs), like A_2A_R and A_2B_R [127]. Adenosine receptors are pleiotropic, and other G-protein subtypes (G_o_, G_q_, G_olf_) are also involved in signal transduction depending on the degree of activation or cellular/subcellular localization [128]. In addition to the exofacial expression of adenosine receptors, adenosine availability and extracellular concentration are also crucial in distinguishing which adenosine receptor subtype is activated. Interstitial adenosine levels increase under conditions of high metabolic demand, such as exercise, and low energy intake, such as ischemia, to reach physiologically relevant concentrations. Adenosine is released into the extracellular space to restore the balance between local energy needs and energy supply [129].

The transfer of adenosine to both sides of the cell is performed via specific proteins, called nucleoside transporters (NTs). NTs alter cellular and plasma adenosine levels [130,131]. The transport of adenosine across the cell membrane is crucial because it regulates the levels of extracellular adenosine that come into contact with surface receptors. Two types of NTs were identified: (i) four equilibrative nucleoside transporters (ENT1 to ENT4) [132,133]; and (ii) three concentrative transporters (CNT1 to CNT3) [134]. The increase in ENT and CNT activities may reduce the availability of extracellular adenosine for its receptors, which attenuates their effects. Therefore, NTs act as crucial players in adenosine function by controlling the local levels of adenosine near adenosine receptors. The effectiveness of this transport system is particularly active in humans, and it is responsible for the extremely short half-life of adenosine in human blood. Adenosine, its receptors and nucleoside transporters together form the “adenosinergic system”, which exerts fine regulation in multiple physiological and pathophysiological processes [135,136,137].

### 4.1. Adenosine Receptors in the Immune System

Adenosine and its receptors play a role in the modulation of inflammation and the immune response [138]. Adenosine released into extracellular spaces occurs in response to inflammation [139,140]. Molecular agents other than adenosine stimulate (agonist) or inhibit (antagonist) its receptors in certain diseases. Coffee consumption is associated with a lower risk of type 2 diabetes. Overall, the experimental and epidemiological evidence elucidated a protective effect of coffee consumption in this disease and showed that caffeine reduced the production of pro-inflammatory cytokines and increased anti-inflammatory processes via the inhibition of A_1_R, A_2A_R and A_2B_R signaling. Caffeine produces biphasic effects. Effects of low doses of caffeine are mediated by adenosine blockade. High dose effects are not due to adenosine antagonism but have a less well known underlying mechanism [141,142]. Coffee consumption seems to have beneficial effects on subclinical inflammation and HDL cholesterol and repeated intake of caffeine paradoxically leads to upregulation of A_2A_R which is accompanied by sensitization to the actions of the agonist HE-NECA [143,144].

Pharmacological activation of A_2A_R using the agonist ATL-146e in diabetic rats ameliorated diabetes-induced histological and functional changes in kidneys and reduced the inflammation associated with diabetic nephropathy [145]. Chronic treatment with the A_2A_R agonist CGS-21680 prevented proteinuria and glomerular damage in diabetic rats via an anti-inflammatory mechanism that was independent of oxidative stress and kidney hypoxia [146]. Adenosine receptors regulate the immune system and act on inflammation and immunes disorders. A_2A_R agonists have anti-inflammatory actions in numerous diseases, including ischemia, arthritis, sepsis, pulmonary and bowel disease and wound healing. Studies in mice and rats demonstrated that the anti-inflammatory effects of methotrexate were lost when animals were treated with A_2A_R and A_3_R antagonists and when these receptors were deleted. Similar effects were observed in patients with rheumatoid arthritis treated with methotrexate who ingested large amounts of caffeine, an adenosine receptor antagonist [138,147]. Activation of A_2A_R by endogenous adenosine contributed to the production of interleukin-10 (IL-10) in polymicrobial sepsis [148]. IL-10-mediated signaling was significantly attenuated in macrophages derived from A_2B_R knockout mice [149]. Accumulating evidence suggests that chronic silent inflammation is a key feature of abdominal obesity, metabolic syndrome, type 2 diabetes and cardiovascular disease [150,151,152]. The contribution of inflammation to the disease is supported by the results of preclinical studies and new clinical trials using anti-inflammatory approaches [151]. A_3_R has a complex immune-modulatory role and may promote pro- and anti-inflammatory processes. Current knowledge on the role of adenosine A_3_R in CVD is limited, but studies in knockout mice demonstrated that the absence of A_3_R signaling prevented the development of hypertension and alleviated kidney and cardiovascular damage via various mechanisms, including a reduction in the population of antigen-presenting cells, an increase in immune homeostasis and a decrease in chronic inflammation and oxidative stress during disease [153]. Activation of A_2A_R and A_2B_R via the inhibition of oxidative activity may have strong modulatory effects on the immune system and arrest the progression of adverse effects in various disorders.

Neutrophils express the four types of adenosine receptors [154]. A_2A_R has a high affinity for adenosine, and it is expressed on basophils, mast cells, monocytes, dendritic cells, T and B cells and NK cells [155]. Several studies demonstrated that activation of A_2A_R signaling enhanced IL-10 production, inhibited macrophage infiltration, suppressed pro-inflammatory cytokines from T cells and myeloid cells, and increased regulatory T cell expression [147]. Suppressive effects of the A_2A_R on leukocyte function in vitro and in vivo are widely described [156]. A_2A_R agonists inhibit human neutrophil activation [157,158,159] and reduce cytokine production induced by T cell receptor engagement [160]. A_2A_R agonists inhibit neutrophil adhesion and infiltration, inflammatory cytokine production, neutrophil degranulation and oxidative burst [156]. A_2A_R knockout mice exhibited increased leukocyte migration and poor defense against tissue damage in various models of inflammation, such as lipopolysaccharide-induced lung damage, inflammation by methotrexate and its analog MX-68 and the formation of excisional wounds [161,162,163,164]. In mice deficient in A_2A_R and apolipoprotein E, which play crucial roles in the regulation of lipid metabolism and atherogenesis, the absence of A_2A_R improved leukocyte recruitment and increased the size of the arterial neointima in injured carotid arteries [165]. Neutrophil A_2A_R inhibited inflammation in a rat model of meningitis and the oxidative activity of human neutrophils via the AMP/PKA cyclic pathway [166,167]. A_2A_R are negative immune regulators that may be used to manipulate T cells, particularly during anti-tumor immune responses [168].

### 4.2. The Adenosinergic and Cardiovascular Systems

The adenosinergic system regulates many physiological and pathophysiological states via modification of adenosine production or the tissue expression of different types of receptors [169,170]. During myocardial ischemia, most cell types, including myocytes and vascular endothelial cells, release adenosine extracellularly into the blood in response to decreased oxygen levels. The adenosine concentration in the coronary sinus is proportional to the degree of coronary artery stenosis [171]. Elevated adenosine plasma levels are higher in severe CAD than in healthy subjects [172]. Notably, HHCy was also associated with severe CAD [173]. 

Most of the cells involved in the cardiovascular system express adenosine receptors on their surface [174]. A_1_R, A_2A_R, A_2_B and A_3_R were localized in the heart, and their distribution depends on the tissue [175]. For example, high levels of A_1_R, with a high affinity for adenosine, are expressed in the atria, primarily in the right atrium and lower expression was found in ventricular myocytes [174,176]. A_1_R is also expressed in smooth muscles and endothelial coronary tissues [177]. A_2A_R is fully expressed in the cardiovascular system, particularly in the arteries, atria and ventricular tissue [174,178,179].

Adenosine and its receptors strongly affect heart rhythm and blood pressure [169]. A_1_R and A_3_R protect the cardiovascular system against ischemia/reperfusion injury by improving mitochondrial function [140]. During spontaneous or induced myocardial ischemia, adenosine acts on coronary blood flow via A_2A_R and A_2B_R. During ischemia, adenosine release from endothelial cells induces A_2A_R activation and cAMP production, which correlate with coronary vasodilation [180]. Myocardial anti-ischemic properties were attributed to A_1_R and A_2A_R because a selective A_1_R agonist restricted the increase in heart rhythm and A_2A_R knockout mice suffer from tachycardia and hypertension [181]. In addition to this cardioprotective property, the activation of A_2A_R exerts a pleiotropic action on coronary smooth muscle cells, endothelial cells and mononuclear cells, which leads to vasodilation, neoangiogenesis and decreased levels of pro-inflammatory cytokines (IL-1β, IL-6 and TNF-α) [127,182]. A_2B_R is also involved in coronary vasodilation in myocardial ischemia [183,184]. Although the activation of A_2A_R and A_2B_R upon exposure to high plasma levels of adenosine results in beneficial effects on the myocardium for a short-term period, the long-term activation of these receptors during chronic exposure to high adenosine plasma levels is harmful [127]. Primarily A_2A_R, but also A_2B_R despite having the lowest affinity for adenosine, are expressed in ventricular myocytes and fibroblasts and modulate inotropic properties and ventricular function in animals [185,186]. Reported evidence reveals that activation of A_2B_R in smooth muscles of coronary arteries contributes to coronary vasodilation [187]. The myocardial expression of A_3_R is very low, but it is expressed within the heart and seems to play a role in coronary artery muscle cells and other smooth muscle cells, where these receptors modulate inward potassium channels and cAMP production [188,189,190].

### 4.3. Adenosine and Its Receptors in Coronary Artery Disease (CAD)

Acute coronary syndrome and its most common consequence, sudden cardiac death, is a major public health problem and accounts for approximately 50% of all cardiovascular deaths, of which at least 25% are the first symptomatic cardiac events [191]. Coronary artery stenosis causes a severe imbalance in oxygen supply and demand, which results in ischemia. Reperfusion strategies are the current standard treatment for acute coronary syndrome, but these strategies may lead to paradoxical cardiomyocyte dysfunction, known as ischemic reperfusion injury, and the exact mechanisms are not known (e.g., deep inflammatory response, neurohumoral activation and oxidative stress) [192]. Adenosine exerts numerous effects in the heart, including modulation of the cardiac response to stress, particularly during myocardial ischemia and reperfusion [193]. Adenosine is also a potent autocrine and paracrine immunosuppressive nucleoside that is released in the vicinity of damaged cells in conditions of metabolic stress, such as ischemia, tissue injury or inflammation [155,194]. Extracellular adenosine has been referred to as a “safety signal” that dampens hypoxia-induced inflammation during ischemia and reperfusion [195]. Extracellular conversion of ATP to adenosine has a central role in attenuating sterile inflammation during ischemia-reperfusion injury. Experimental studies showed that pharmacological strategies to increase the breakdown of ATP to adenosine were effective in attenuating tissue injury and pathogen-free inflammation during ischemia and reperfusion [196,197,198,199,200,201,202]. Several experimental trials provide evidence of a protective role of adenosine signaling in models of ischemia and reperfusion via activation of A_2A_R on inflammatory cells [203,204,205] or activation of A_2B_R on the vascular endothelium, epithelium or cardiac myocytes [200,206,207,208]. A_2B_R signaling controls the expression of the circadian protein Per2, which stabilizes hypoxia-inducible factor (HIF), promotes glycolytic metabolism and has cardioprotective effects. Exposure of mice to intense light stabilized Per2 in the heart and reduced cardiac injury after myocardial ischemia [209]. Activation of A_2A_R on T cells attenuated ischemia and reperfusion in experimental models of sickle cell disease [204]. A_2A_R reduces inflammation by acting on pro-inflammatory cells to attenuate the release of pro-inflammatory cytokines and decreasing the level of endothelial adhesion molecules. Numerous preclinical studies using A_2A_R agonists and antagonists, A_2A_R knockout and chimeric mice showed the therapeutic potential of A_2A_R agonists for the treatment of ischemia-reperfusion injury and autoimmune diseases [210].

We showed that patients with CAD had low levels of A_2A_R on the surface of peripheral blood mononuclear cells (PBMCs) [129,172,211]. PBMCs are a valuable surrogate for cardiovascular cells to study the adenosinergic profile in patients with CAD because the behavior of A_2A_R in the two cell types is similar in terms of A_2A_R level and cAMP production, which was reported for the left ventricle in cardiac transplant recipients, the aorta and coronary artery tissues, and femoral arteries [172,212,213]. The properties of A_2A_R expressed by PBMCs mirror the properties of A_2A_R in the vascular wall likely because PBMCs are exposed to blood flow and are in contact with all tissues. This correlation of expression and function of A_2A_R in the two compartments is certainly a reflection of systemic regulation. Therefore, it offers a unique opportunity to study the adenosinergic system and its behavior under ischemic conditions in coronary arteries [126].

A recent study correlated a low level of A_2A_R in PBMCs with a high level of plasma cholesterol in patients with familial hypercholesterolemia, which reinforces the link between the immunosuppressive adenosinergic system and chronic inflammation in the atherogenic process [214]. Notably, the final product of adenosine degradation, uric acid, was significantly associated with CAD and endothelial dysfunctions [215,216]. The decrease in the level and activity of A_2A_R contributes to the maintenance and worsening of CAD via modification of the adaptive vasodilation of the coronary arteries when an oxygen supply is necessary, such as during the exercise stress test [129,136].

## 5. Adenosinergic System, HCy and H_2_S in CAD

As previously shown, adenosine comes from the methionine cycle, and its production is linked to HCy metabolism. HCy and adenosine are independently associated with cardiovascular disorders. Recent data suggest a link between HCy and adenosine, which may explain the higher cardiovascular risk observed in HHCy. For example, hypoxia increased intracellular adenosine production from ATP to form SAH with HCy via SAH hydrolase. HCy enters the cell through transporters, and it is no longer the limiting substrate for the production, along with adenosine, of SAH by reversing the reaction of SAH hydrolase. The resulting high concentration of SAH forces the enzyme to return, as in basal conditions, to the production of adenosine and HCy, which accumulate in the cell and are catabolized to uric acid and H_2_S, respectively [217]. Notably, HCy plasma levels in CAD patients correlated with adenosine and uric acid plasma levels and a decrease in A_2A_R production and function [218,219]. Alternately, adenosine induced a time- and dose-dependent increase in HCy in hepatoma cultured cells [218]. H_2_S also decreased the level of A_2A_R expression in lymphocytes, which led to adenosinergic immunosuppression and the promotion of inflammation against a background of elevated HCy [217]. The downregulation of A_2A_R due to the increase of HCy in the blood may be explained by a decrease in the level of A_2A_R in the PBMCs, which was found concomitant with their accumulation in extracellular vesicles isolated from the plasma of CAD patients with HHCy [220]. A_2A_R is the major adenosine receptor expressed in platelets and mediates the inhibition of platelet aggregation [221]. A decrease in the expression of A_2A_R on platelets due to HHCy via H_2_S could promote their aggregation. Consistently, HHCy in patients was associated with increased platelet aggregation via the H_2_S pathway, which contributed to atherothrombosis, stroke and myocardial infarction [86]. Therefore, we propose that HHCy, via the elevated production of H_2_S, damages the cardiovascular system by reducing the number of A_2A_R expressed on cardiac myocytes, and endothelial and immune cells under ischemia-hypoxia (Figure 4). By neutralizing vasodilation and the adenosinergic immune suppression via the action of H_2_S on NF-κB and HIF, HHCy may decrease blood flow and exacerbate T cell infiltration and the concomitant release of pro-inflammatory cytokines in cardiovascular tissue. Therefore, the increased risk of coronary heart disease associated with HHCy may be the consequence of elevated levels of H_2_S on the adenosinergic system.

## 6. Conclusions and Future Directions

This article is an overview of the available evidence, which may appear partial and/or controversial, on the interaction between the adenosinergic system and the metabolism of HCy. The way in which these interactions are orchestrated in the cardiovascular system, particularly under conditions such as inflammation or hypoxia/ischemia, has highlighted the putative role of H_2_S as a gas mediator in various immune/inflammatory diseases affecting the cardiovascular system, with CAD being the most characterized. These data are consistent with the hypothesis that HCy participates in CAD pathophysiology by lowering A_2A_R expression on blood vessels and T cells to reduce coronary blood flow and promote inflammation, respectively.

This described system may be a contributor, but given all of the side effects of HHCy on CVD and the compelling evidence that H_2_S protects against CAD, it is possible that it all depends on the context and cell type. Many other factors, such as age, weight, lipid status and a family history of high blood pressure or diabetes mellitus may also interact. The question of the importance of the production of H_2_S in situ or in the peripheral circulation in relation to the level of expression of transsulfuration enzymes in the cells of the cardiovascular and immune systems remains to be studied. However, measuring the level of A_2A_R expression on PBMCs could offer a new risk factor for CVD. Innovative and targeted treatments on the modulation of A_2A_R by H_2_S could be considered.

## Figures and Tables

**Figure 1 ijms-22-01690-f001:**
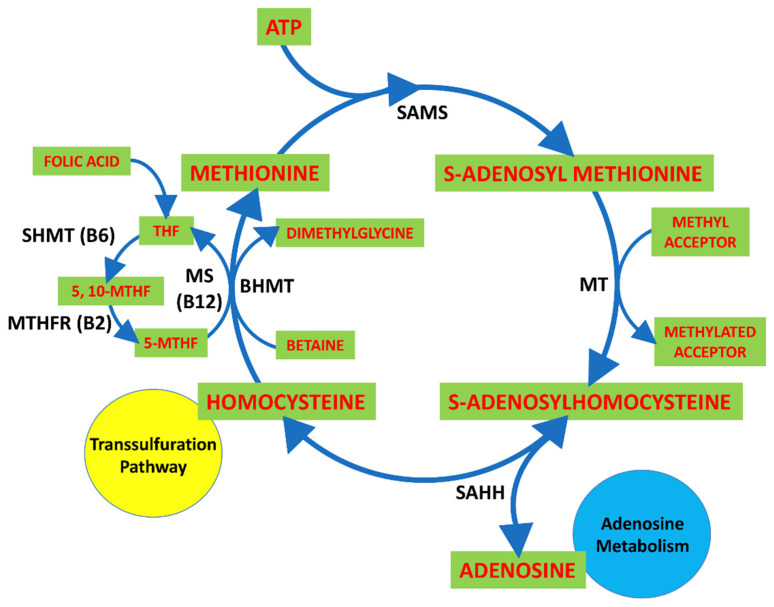
Methionine cycle. Homocysteine (HCy) is biosynthesized from methionine by S- adenosylmethionine synthetase (SAMS), methyltransferase (MT) and S-adenosylhomocysteine hydrolase (SAHH) in sequential steps. Methionine is activated by condensation with adenosine triphosphate (ATP) to yield the ubiquitous methyl donor SAM, which is transformed into S-adenosylhomocysteine (SAH) by donating its methyl group to the substrates of methylation reactions. SAH gives rise to HCy in a reversible reaction that favors SAH over HCy production. SAH is a competitive inhibitor of methylation reactions, and rapid elimination of adenosine and HCy is required to prevent its accumulation. HCy may be remethylated to methionine by methionine synthase (MS), which requires folate and vitamin B12, and betaine homocysteine S-methyltransferase (BHMT), which requires betaine, a metabolite of choline. Remethylation of HCy via MS requires 5-methyltetrahydrofolate (5-MTHF), which is derived from 5,10-methylenetetrahydrofolate (5,10-MTHF) in a reaction catalyzed by MTHFR with vitamin B2 as a cofactor. 5-MTHF is converted into tetrahydrofolate (THF) after it donates its methyl group, and THF is converted into 5,10-MTHF by serine hydroxymethyltransferase (SHMT) with vitamin B6 as a cofactor to complete the folate cycle. HCy may enter the transsulfuration pathway, and adenosine can reach its metabolic process.

**Figure 2 ijms-22-01690-f002:**
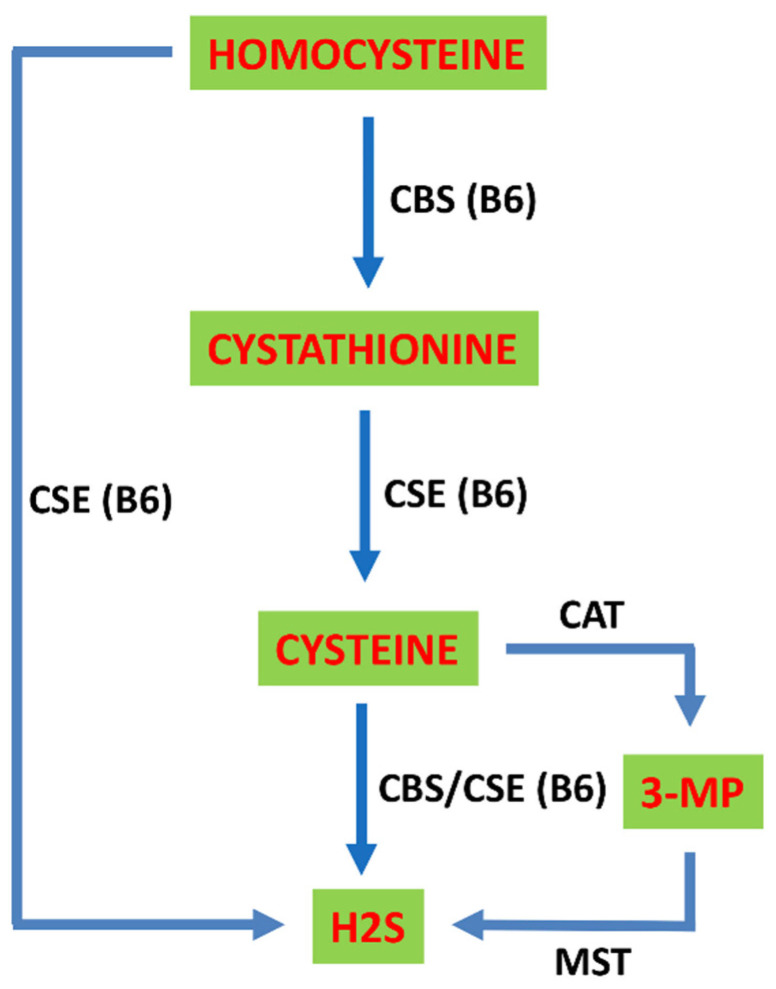
Transsulfuration pathway. HCy may be sequentially converted into cystathionine then cysteine by two vitamin B6-dependent enzymes, cystathionine β-synthase (CBS) and cystathionine γ-lyase (CSE), which subsequently results in the generation of H_2_S. HCy and cysteine are substrates for H_2_S production by CBS, CSE, cysteine aminotransferase (CAT) and 3-mercaptopyruvate (3-MP) sulfurtransferase (MST).

**Figure 3 ijms-22-01690-f003:**
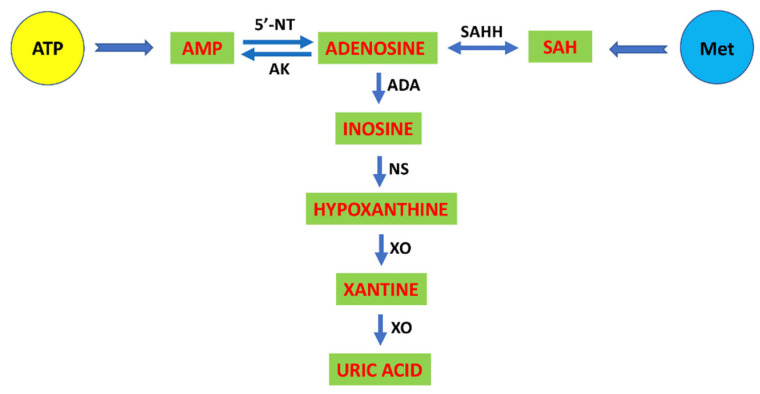
Intracellular adenosine metabolism. Adenosine comes primarily from ATP degradation in the intra- and extracellular environment. In the cytosol, adenosine produced from AMP via 5′-nucleotidase (5′-NT) may be phosphorylated again by adenosine kinase (AK). Some adenosine arises intracellularly from the metabolism of methionine (Met) via the reversible action of S-adenosylhomocysteine hydrolase (SAHH). Intra- and extracellular adenosine is successively degraded into inosine by adenosine deaminase (ADA), into hypoxanthine by nucleosidase (NS) and into xanthine then uric acid by xanthine oxidase (XO).

**Figure 4 ijms-22-01690-f004:**
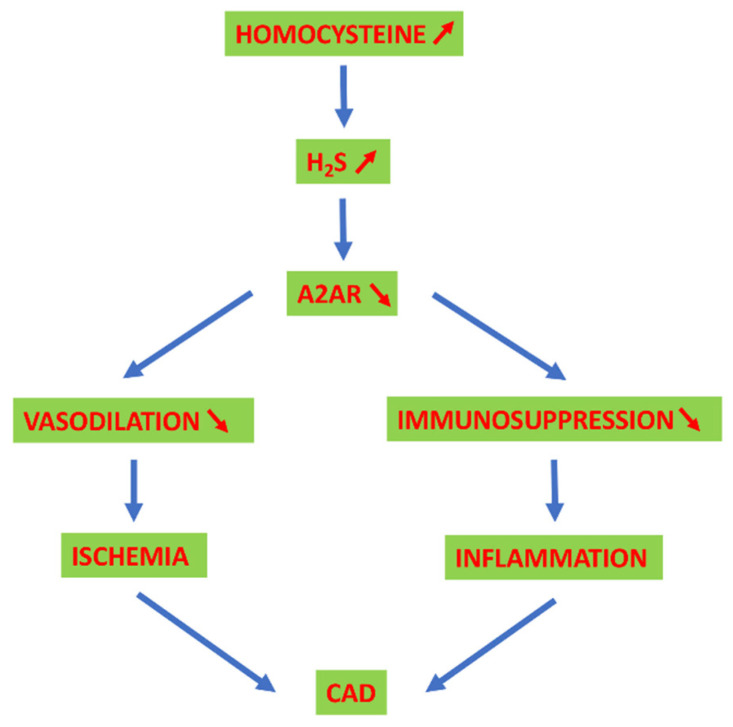
Proposed mechanism of the role of HHCy in CAD. In cases of CAD with HHCy, H_2_S accumulates in cells and reverses adenosine-induced A_2A_R expression by endothelial and immune cells via the NF-κB pathway. Therefore, the vasodilation of the arteries and the immunosuppressive action of the lymphocytes are hampered, which promotes the processes of ischemia and inflammation, respectively, that aggravate CAD.

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
