# Peer review of "Hyperhomocysteinemia and Cardiovascular Disease: Is the Adenosinergic System the Missing Link?"

_ijms, 2021, doi:10.3390/ijms22041690_

Round 1

Reviewer 1 Report

The review related to hyperhomocysteinemia and cardiovascular disease is focused on the adenosinergic system as the potential link. This manuscript described the role of homocysteine and the products of homocysteine metabolism in the development of cardiovascular diseases. The final part is focused on the role of adenosine in the immune system concerning for cardiovascular diseases. This topic is important and the paper is well-written.

I have several comments:

The authors should revise the first part as the Introduction. Specifically, it should be included clear definition and pathomechanisms of hypertension and atherosclerosis. Moreover, the aims of this review are missing. 

I recommend including future directions in the part Conclusion.

p.9, line 341 – please specify the kind of anti-inflammatory cytokines.

I recommend revising the references focusing on the last studies (from 5-7 years).     

Author Response

The review related to hyperhomocysteinemia and cardiovascular disease is focused on the adenosinergic system as the potential link. This manuscript described the role of homocysteine and the products of homocysteine metabolism in the development of cardiovascular diseases. The final part is focused on the role of adenosine in the immune system concerning for cardiovascular diseases. This topic is important and the paper is well-written.

We thank reviewer 1 for his commentary on the importance of the topic and the writing of the review as well as for his constructive criticism which allows us to improve the journal.

I have several comments:

The authors should revise the first part as the Introduction. Specifically, it should be included clear definition and pathomechanisms of hypertension and atherosclerosis. Moreover, the aims of this review are missing. 

We renamed Part 1 “Introduction” and indicated the mechanism of atherosclerotic plaque formation (lines 33-42), the link between hypertension and high cholesterol (lines 50-54), and the purpose of this review (lines 56 to 60).

I recommend including future directions in the part Conclusion.

We have added a second paragraph to part 6 regarding future directions (lines 479-487).

p.9, line 341 – please specify the kind of anti-inflammatory cytokines.

The 3 main anti-inflammatory cytokines are now indicated (lines 367, 368).

I recommend revising the references focusing on the last studies (from 5-7 years). 

We have already selected the most relevant references but there are important references that are over 7 years old and we think they are worth mentioning. In addition, 4 “older” references have been deleted to add new references in accordance with comments from other reviewers.

Reviewer 2 Report

This is a very-well written review dedicated to the role of adenosinergic system in the mechanism of hyperhomocysteinemia-related cardiovascular disorders. The authors have focused on the homocysteine (Hcy) degradation product hydrogen sulfide as possible mediator of cell damage. The manuscript is well organized and quality of presentation is very good. Presented views can be interesting for researchers in the field of cardiovascular disease. There are few minor concerns which should be attended to.

Lines 263-277: literature data on the protective role of coffee consumption related to its inhibitory effect on adenosine receptors (ARs) are presented together with data showing protective effects of AR agonists ATL-146e and CGS-21680. This paradoxical effect of coffee consumption should be at least briefly explained.

Although the review is focused on the role of hydrogen sulfide as mediator of HHcy effect on adenosinergic system, other possible mechanisms, such as altered DNA methylation resulting in changes of ARs expression and Hcy-induced oxidative damage to ARs genes/proteins, should be mentioned.

Line 414: “…. adenosine and H2S, which accumulates …” should be …. adenosine and Hcy, which accumulate …

Author Response

This is a very-well written review dedicated to the role of adenosinergic system in the mechanism of hyperhomocysteinemia-related cardiovascular disorders. The authors have focused on the homocysteine (Hcy) degradation product hydrogen sulfide as possible mediator of cell damage. The manuscript is well organized and quality of presentation is very good. Presented views can be interesting for researchers in the field of cardiovascular disease. There are few minor concerns which should be attended to.

We thank reviewer 1 for his comments on the quality of the organization and presentation of the review as well as for his constructive criticism which enables us to improve it.

Lines 263-277: literature data on the protective role of coffee consumption related to its inhibitory effect on adenosine receptors (ARs) are presented together with data showing protective effects of AR agonists ATL-146e and CGS-21680. This paradoxical effect of coffee consumption should be at least briefly explained.

The effect of caffeine is complex and an attempted explanation has been given (lines 286-292). For this, the old reference 144 has been replaced by a more appropriate one.

Varani K, Portaluppi F, Gessi S, Merighi S, Ongini E, Belardinelli L, Borea PA. Dose and time effects of caffeine intake on human platelet adenosine A(2A) receptors : functional and biochemical aspects.

.Circulation. 2000 Jul 18;102(3):285-9. doi: 10.1161/01.cir.102.3.285.

Although the review is focused on the role of hydrogen sulfide as mediator of HHcy effect on adenosinergic system, other possible mechanisms, such as altered DNA methylation resulting in changes of ARs expression and Hcy-induced oxidative damage to ARs genes/proteins, should be mentioned.

The effect of hyperhomocysteinemia on ROS production has already been mentioned with several references (references 37-40). Its action on DNA methylation has been added (lines 121, 122). On this occasion, 2 old references (41, 42) have been replaced by 2 new ones.

Castro,R.;Rivera,I.;Struys,E.A.;Jansen,E.E.W.;Ravasco,P.;Camilo,M.E.;Blom,H.J.;Jakobs,C.;Tavaresde Almeida, I. Increased homocysteine and S-adenosylhomocysteine concentrations and DNA hypomethylation in vascular disease. Clin. Chem. 2003, 49, 1292–1296.

Choumenkovitch, S.F.; Selhub, J.; Bagley, P.J.; Maeda, N.; Nadeau, M.R.; Smith, D.E.; Choi, S.-W. In the cystathionine beta-synthase knockout mouse, elevations in total plasma homocysteine increase tissue S-adenosylhomocysteine, but responses of S-adenosylmethionine and DNA methylation are tissue specific. J. Nutr. 2002, 132, 2157–2160.

We have also indicated the role of H2S on epigenetics and replaced the old reference 102, which was not relevant, with a new more appropriate one.

Wang Y, Yu R, Wu L, Yang G. Hydrogen sulfide signaling in regulation of cell behaviors. Nitric Oxide. 2020 Oct 1;103:9-19.

Line 414: “…. adenosine and H2S, which accumulates …” should be …. adenosine and Hcy, which accumulate …

The sentence has been corrected on line 440.

Reviewer 3 Report

This is a wonderfully well written and polished review paper covering aspects of hyperhomocysteinemia, hydrogen sulfide, and the adenosinergic system. We are still learning much about the biological role of hydrogen sulfide, particularly instances where it can act as an anti-inflammatory agent, and other where it can be proinflammatory. These situations certainly depend on context, H2S concentration, and the organ / cell type under investigation. In section 5, the authors propose a mechanism whereby hyperhomocysteinemia actually drives an increase in H2S levels, which then suppresses A2AR signaling, ultimately leading to coronary artery disease. As the authors allude to, this is likely to be received as controversial. The proposed mechanism is interesting and well-reasoned, though is likely dependent on context and cell type. I have a few recommendations to improve the quality of the manuscript.

1. Line 183 – Reference [102] does not appear to be the correct reference for this statement. It deals with H2S and ONOO- in neuroblastoma cells, not atherosclerosis.

2. Line 430 – I believe “cardiovascular cells” is too broad of a cell type to include in the hypothesis proposal. Cardiovascular implies all cell types in the heart, endothelium, arterial smooth muscle, veins, etc. There is ample evidence that HHCy coincides with reduced H2S production in cardiac myocytes and endothelial cells, as well as evidence that HCy inhibits CSE activity in these cells.

3. Line 435 – I have issue with the use of “primarily” here. This system described may be a contributor but given all of the adverse effects of HHCy CVD develop and the overwhelming evidence that H2S is protective against coronary heart disease, the word “primarily” in this context is an overstatement.

Author Response

This is a wonderfully well written and polished review paper covering aspects of hyperhomocysteinemia, hydrogen sulfide, and the adenosinergic system. We are still learning much about the biological role of hydrogen sulfide, particularly instances where it can act as an anti-inflammatory agent, and other where it can be proinflammatory. These situations certainly depend on context, H2S concentration, and the organ / cell type under investigation. In section 5, the authors propose a mechanism whereby hyperhomocysteinemia actually drives an increase in H2S levels, which then suppresses A2AR signaling, ultimately leading to coronary artery disease. As the authors allude to, this is likely to be received as controversial. The proposed mechanism is interesting and well-reasoned, though is likely dependent on context and cell type. I have a few recommendations to improve the quality of the manuscript.

We thank reviewer 3 for their complimentary comments and constructive advice.

  1. Line 183 – Reference [102] does not appear to be the correct reference for this statement. It deals with H2S and ONOO- in neuroblastoma cells, not atherosclerosis.

Sorry, this is not the correct reference. We took advantage of a response to review 2 to replace it with another more appropriate reference. 

  1. Line 430 – I believe “cardiovascular cells” is too broad of a cell type to include in the hypothesis proposal. Cardiovascular implies all cell types in the heart, endothelium, arterial smooth muscle, veins, etc. There is ample evidence that HHCy coincides with reduced H2S production in cardiac myocytes and endothelial cells, as well as evidence that HCy inhibits CSE activity in these cells.

 The sentence has been corrected on line 456.

  1. Line 435 – I have issue with the use of “primarily” here. This system described may be a contributor but given all of the adverse effects of HHCy CVD develop and the overwhelming evidence that H2S is protective against coronary heart disease, the word “primarily” in this context is an overstatement.

We have deleted “primarily” from line 461.

Round 2

Reviewer 1 Report

In lines 367,368 the cytokines IL-1beta, IL-6, and TNF-alpha are not the main anti-inflammatory cytokines, they represent proinflammatory cytokines. Thus, I am not sure of the correct citation. The authors have to correct it.

Author Response

The critic is absolutely right. IL1-beta, IL-6 and TNF-alpha are indeed pro-inflammatory. We apologize for this error and thank the reviewer for pointing it out. Reference 127 is correct. This is a recent review by Borea et al. which refers to various publications on the subject. But we transcribed the data incorrectly and the words reversed. In fact, activation of A2AR decreases the production of pro-inflammatory cytokines. The sentence (line 367,368) is now corrected. Reference 182 relates more precisely to the important action in cardiology of A2AR on vasodilation.

This manuscript is a resubmission of an earlier submission. The following is a list of the peer review reports and author responses from that submission.